# The Role of TAM Receptors in Bone

**DOI:** 10.3390/ijms25010233

**Published:** 2023-12-23

**Authors:** Janik Engelmann, Deniz Ragipoglu, Isabel Ben-Batalla, Sonja Loges

**Affiliations:** 1Department of Oncology, Hematology and Bone Marrow Transplantation with Section Pneumology, Hubertus Wald Comprehensive Cancer Center Hamburg, University Medical Center Hamburg-Eppendorf, 20246 Hamburg, Germany; ja.engelmann@uke.de; 2Department of Tumor Biology, Center of Experimental Medicine, University Medical Center Hamburg-Eppendorf, 20246 Hamburg, Germany; 3DKFZ-Hector Cancer Institute at the University Medical Center Mannheim, 68167 Mannheim, Germany; deniz.ragipoglu@dkfz-heidelberg.de (D.R.); isabel.benbatalla@dkfz-heidelberg.de (I.B.-B.); 4Division of Personalized Medical Oncology (A420), German Cancer Research Center (DKFZ), German Center for Lung Research (DZL), 69120 Heidelberg, Germany; 5Department of Personalized Oncology, University Hospital Mannheim, Medical Faculty Mannheim, University of Heidelberg, 68167 Mannheim, Germany

**Keywords:** TYRO3, AXL, MERTK, PROS1, GAS6, TAM receptors, osteoblasts, osteoclasts, bone, cancer, bone metastasis, multiple myeloma

## Abstract

The TAM (TYRO3, MERTK, and AXL) family of receptor tyrosine kinases are pleiotropic regulators of adult tissue homeostasis maintaining organ integrity and self-renewal. Disruption of their homeostatic balance fosters pathological conditions like autoinflammatory or degenerative diseases including rheumatoid arthritis, lupus erythematodes, or liver fibrosis. Moreover, TAM receptors exhibit prominent cell-transforming properties, promoting tumor progression, metastasis, and therapy resistance in various cancer entities. Emerging evidence shows that TAM receptors are involved in bone homeostasis by regulating osteoblastic bone formation and osteoclastic bone resorption. Therefore, TAM receptors emerge as new key players of the regulatory cytokine network of osteoblasts and osteoclasts and represent accessible targets for pharmacologic therapy for a broad set of different bone diseases, including primary and metastatic bone tumors, rheumatoid arthritis, or osteoporosis.

## 1. Introduction

TYRO3, AXL, and MERTK are members of the TAM family of receptor tyrosine kinases and are widely expressed throughout the body [1]. They share a similar structure comprised of an extracellular domain containing two immunoglobulin (Ig)-like repeats and two fibronectin type III (FNIII) repeats. The extracellular domain, which allows receptor–ligand interaction, is followed by a transmembrane domain and a cytoplasmic tyrosine kinase domain [2]. The most studied TAM receptor ligands are growth arrest-specific 6 (GAS6) and protein S (PROS1), which share similar structural motifs and are vitamin K-dependent [3]. Both GAS6 and PROS1 comprise three main elements: an N-terminal γ-carboxyglutamic acid (Gla) domain, which regulates Ca^2+^-dependent binding to phosphatidylserine (PtdSer); four epidermal growth factor (EGF)-like repeats; and a C-terminal sex hormone-binding globulin (SHBG) domain [4,5]. The SHBG domain mediates receptor–ligand binding through its two laminin G-like (LG) domains.

All three TAM receptors are activated by GAS6, which binds receptors with different affinities (AXL > TXRO3 > MERTK), whereas PROS1 activates TYRO3 and MERTK but not AXL [5,6]. TAM receptors are activated by binding of the ligand LG domains to Ig-like domains of the receptor. Optimal receptor–ligand interaction requires the presence of Ca^2+^ and PtdSer-presenting membrane (i.e., apoptotic cells or enveloped virus) [5]. Activation of the receptor results in receptor dimerization, autophosphorylation of kinase domains, and activation of intracellular signaling cascades [2]. TAM receptors mediate different cellular processes including growth, survival, differentiation, adhesion, migration, and apoptosis [1,2,4].

TAM receptors play a crucial role in physiological tissue homeostasis and inflammation by mediating the clearance of apoptotic cells (such as aged or damaged cells) by professional phagocytes or epithelial cells [7,8,9,10,11,12,13]. Apoptotic cells expose PtdSer on their outer surface, which can bind to TAM receptor ligands [14]. Thereby, GAS6 and PROS1 can function as bridging molecules between phagocytes and apoptotic cells and induce TAM receptor activation resulting in phagocytosis [2]. The process of apoptotic cell engulfment is called efferocytosis, which is a crucial step for the resolution of inflammation [14]. Besides their physiological functions, TAM receptors and their ligands have been associated with chronic inflammatory and autoimmune diseases, such as multiple sclerosis, various rheumatic diseases, and systemic lupus erythematosus [11,15,16,17]. Moreover, TAM receptors are frequently overexpressed in human cancers (such as breast, lung, gastric, metastatic colon, and prostate tumors) and are associated with chemo-resistance, metastasis, and poor prognosis [18,19,20,21,22,23,24].

Recent studies have shown that TAM receptors are involved in bone homeostasis, suggesting that they can be therapeutically targeted in bone disorders [25,26,27]. Bone is a mineralized connective tissue that provides mechanical support for the body and protects internal organs [28]. Besides these functions, it serves as a source of cytokines, hormones, and growth factors, which provides an ideal environment for hematopoietic stem cells (HSCs) [29]. Bone tissue is highly dynamic and continuously undergoes remodeling to maintain bone strength and mineral homeostasis through the coordinated action of bone-forming osteoblasts, bone-resorbing osteoclasts, and osteocytes [30]. Osteoblasts derive from bone marrow-resident mesenchymal stem cells (MSCs), which can differentiate into adipocytes, chondrocytes, osteoblasts, and vascular smooth muscle cells. Osteoblastogenesis is mediated by the induction of master transcription factors runt-related transcription factor 2 (RUNX2) and transcription factor Sp7 (OSTERIX) [30]. Mature osteoblasts secrete an unmineralized bone matrix consisting of type I collagen, which is mineralized by the deposition of calcium and hydroxyapatite [31]. In the termination phase, osteoblasts either undergo apoptosis and become bone lining cells or are embedded into the bone matrix and terminally differentiate into residing osteocytes [32]. Osteocytes are the most abundant and long-lived cells in the bone and orchestrate bone remodeling by sensing and transducing mechanical strains into biochemical signals [33]. Osteoclasts are multinucleated cells and derive from mononuclear HSCs, particularly bone marrow monocyte–macrophage precursors [34]. The remodeling process starts with the recruitment and activation of osteoclast precursor cells [30]. Active osteoclasts attach to the bone and degrade the bone matrix through acidification and the production of enzymes like tartrate-resistant acid phosphatase (TRAP), cathepsin K, or matrix metalloproteinases [32,34]. Following bone resorption, osteoclasts undergo apoptosis and osteoblasts deposit new bone before they become bone lining cells or osteocytes [30]. The fine equilibrium between bone resorption and bone formation is crucial as excessive bone resorption by osteoclasts with insufficient newly formed bone causes osteoporosis. On the contrary, excessive bone formation versus resorption may result in osteopetrosis [32]. The physiological balance in bone remodeling is regulated by different local and systemic factors including biomechanical load, hormones, cytokines, and chemokines [30,32]. In this review, we will focus on the role of TAM receptors and their ligands in bone under physiological and pathological conditions.

## 2. Role of TAM Receptors in Bone Remodeling

### 2.1. TAM Receptors in Osteoclasts

Key molecular signals of osteoclastogenesis are mediated by the macrophage colony-stimulating factor (M-CSF) and the receptor activator of NF-κB ligand (RANKL). In early osteoclast differentiation, M-CSF activates mitogen-activated protein kinase (MAPK)/extracellular signal-regulated kinase (ERK) and phosphoinositide 3-kinases (PI3K)/protein kinase B (AKT) pathways, promoting proliferation, survival, and differentiation. RANKL promotes MAPK/p38/c-Jun N-terminal kinases (JNK), AKT, guanine nucleotide exchange factor VAV3, proto-oncogene tyrosine-protein kinase Src (c-SRC), and tumor necrosis factor receptor-associated factor 6 (TRAF6)/NF-κB signaling cascades to induce microphthalmia-associated transcription factor (MITF), activator protein 1 (AP1), cycling adenosin monophosphate response element binding protein (CREB), and nuclear factor of activated T-cells, cytoplasmic 1 (NFATc1) activation to promote proliferation, survival, motility, adhesion, and differentiation of osteoclast precursors into mature osteoclasts. RANKL signaling can be further strengthened by costimulatory signaling pathways like triggering receptor expressed on myeloid cells 2 (TREM2) or osteoclast-associated receptor (OSCAR) to induce tyro protein tyrosine kinase-binding protein (DAP12)/Fc receptor common γ subunit (FcRγ)-spleen tyrosine kinase (Syk) phospholipase C-(PLCγ) signaling to activate calcium signaling and NFATc1 induction [35].

The role of AXL in osteoclast biology is largely unknown. It was recently shown that AXL is expressed by osteoclast precursors and contributes to osteoclast differentiation in vitro via upregulation of monocyte chemoattractant protein 1 (MCP-1), a regulator of osteoclast cell–cell fusion. Treatment of osteoclast cultures with the small molecule AXL inhibitor Bemcentinib blocked osteoclast differentiation [36]. In contrast, another study showed that treatment of healthy mice with the pan-TAM receptor inhibitor BMS-777607 had no effect on osteoclast numbers on the bone surface [37]. Transgenic animal models with osteoclast-specific knockout are needed to further explore the role of AXL in osteoclast biology.

The first authors showing an implication of TYRO3 in osteoclast differentiation were Nakamura et al. They observed TYRO3 expression in multinucleated osteoclasts in vivo and measured increased bone-resorbing activity of osteoclasts induced by PROS1 and GAS6, whereas osteoclast differentiation was not affected [38]. Another study confirmed that GAS6 did not affect the differentiation or survival of osteoclasts, but stimulated osteoclast bone resorptive function. Mechanistically, they observed that GAS6 induced TYRO3 phosphorylation and ERK signaling, while inhibition of ERK by PD98059 abrogated the GAS6-mediated effect [39]. However, the role of ERK in osteoclast biology is controversial. Germline deletion of Erk1 (Erk1^−/−^) and hematopoietic Erk2 conditional knockout (Mx1-Cre^+^Erk2^flox/flox^) impaired osteoclast formation and function and increased bone mass by decreasing osteoclast formation in vivo [40]. In contrast, a study showed that ERK inhibition increased osteoclastogenesis in RAW 264.7 osteoclast precursor cell line by stimulating AMP-activated protein kinase (AMPK) activation and RANK expression and inhibiting anti-osteoclastogenic factor expression [41]. Therefore, the pro-osteoclastogenic function of ERK might be restricted to early osteoclast precursors. Ruiz-Heiland et al. showed that GAS6 can indeed promote osteoclast differentiation, but these effects were restricted to the presence of low RANKL concentrations. In the absence of RANKL or with high RANKL concentrations, they did not observe the effects of GAS6 on osteoclast differentiation. Ex vivo osteoclast cultures of bone marrow macrophages from Tyro3^−/−^ mice showed decreased osteoclast differentiation and Tyro3^−/−^ mice exhibited increased bone mass and reduced osteoclast numbers in vivo [42].

Although the literature shows a pro-osteoclastogenic role of TYRO3, its biological function appears complex. The pro-osteoclastogenic action of TYRO3 and especially GAS6 in RANKL-mediated osteoclast differentiation seems to be neglectable in vitro. In line with that, another study showed that treatment of osteoclast cultures with soluble TYRO3 led to increased osteoclast differentiation [43]. As soluble TAM receptors serve as extracellular traps for TAM ligands, these results suggest that PROS1 or GAS6 might dampen osteoclastogenesis.

Another study detected increased TYRO3 expression by FACS analysis on the surface of pro-osteoclastogenic CD14^+^CD16^−^ monocyte subpopulation in the blood of rheumatoid arthritis patients, indicating a role for TYRO3 in monocytic osteoclast precursor cells in the circulation [44]. During the development of the bone, osteoclasts derive from erythromyeloid progenitors (EMPs) from the blood islands of the yolk sac. In postnatal life, HSC-derived precursors give rise to osteoclasts [45,46]. Although the anatomical site of the main osteoclast precursor cell pool is still under debate, it is acknowledged that osteoclasts derive mainly from a pool of primed monocytic cells in the circulation. The concept of circulatory osteoclast precursors requires highly dynamic and mobile cells. The migratory mechanisms in systemic circulation and homing into bone spaces are critical points of control for osteoclast formation and bone homeostasis. Entry and exit from the bone marrow cavity are regulated by several chemokines and signaling molecules, such as chemokine (C-X-C motif) ligand (CXCL) 12, CC chemokine ligands (CCL)19, CCL21, CX3CL1, and sphingosine-1-phosphate (S1P) [47,48,49,50]. S1P modulates osteoclast precursor cell migration by two counteracting receptors, circulation-attractive S1P receptor 1 (S1PR1) and bone tropic S1PR2, regulating fine-tuned entrance into and exit from the bones [51,52]. We recently showed that the TAM receptor MERTK might be involved in osteoclast precursor cell homing and motility [25,26]. We demonstrated that myeloid cell-specific deletion of MERTK (LysM-cre^+^Mertk^flox/flox^) increased bone mass and reduced osteoclastogenesis in vivo [26]. Mechanistically, deletion of MERTK in myeloid cells decreased TRAP^+^ mononuclear cells in the bone marrow. Furthermore, osteoclast precursor cells lacking MERTK showed decreased motility via the small GTPase Ras homolog family member A (RHOA), suggesting that MERTK regulates osteoclast precursor cell migration and homing to bone marrow cavity [26]. RHOA is a key regulator of osteoclast differentiation and function. It was found to be responsible for C-C chemokine receptor type 7 (CCR7)-dependent migration and chemotaxis, as well as S1PR2-mediated chemorepulsion of monocytic osteoclast precursors [49,51,53]. In mature osteoclasts, RHOA was found to regulate bone resorptive activity [53,54,55]. The precise role of MERTK and TYRO3 in osteoclast differentiation, migration, and function will be the subject of further studies.

Next to that, TAM receptors play a critical role in macrophages, which share a common hematopoietic origin with osteoclasts and are key effector cells in innate and adaptive immunity [56]. TAM receptors are differentially expressed in macrophage development and maturation into their subsequent subpopulations M1 and M2 [57]. M1 and M2 macrophages can be distinguished by their cytokine profiles, which exert opposite effects on osteoclasts. “Classically activated” M1 macrophages secrete pro-inflammatory cytokines that promote osteoclastogenesis, such as tumor necrosis factor α (TNF-α), interleukin-6 (IL-6), IL-1β, IL-15, and IL-18, while “Alternatively activated” M2 macrophages secrete mostly cytokines, which can inhibit osteoclastic bone resorption such as transforming growth factor β (TGF-β) and IL-10 [58].

However, the influence of M1 vs. M2 macrophage polarization on osteoclastogenesis is under debate. It was shown that RANKL increases M1 macrophages that reside near the growth plate in high proximity to osteoclasts and osteoblasts. Furthermore, treatment with RANKL increased the expression of pro-inflammatory cytokines and inducible nitric oxide synthase (iNOS) in macrophages. The authors suggest that M1 macrophages may finally turn into osteoclasts through induction of RANKL [59]. In contrast, another study showed that in osteoporosis, M2 macrophages but not M1 macrophages differentiate into functional osteoclasts in the presence of RANKL and that estrogen protects M2 macrophages from RANKL stimulation [60]. It has been also shown that M1 macrophages significantly reduced osteoclastogenesis by downregulating master transcription factor NFATc1 and increasing the apoptosis of osteoclasts. This was confirmed in a functional mouse model of ligature-induced periodontitis. M1 and M2 macrophages were adoptively transferred and reduced osteoclastogenesis was observed by the presence of M1 macrophages. This led to the hypothesis that induction of M1 macrophage polarization could be an attractive treatment strategy for reducing osteoclastogenesis in bone diseases [61].

Data on the role of TYRO3 in macrophage polarization are scarce with one study showing that TYRO3 can inhibit lipopolysaccharide (LPS) and interferon-gamma (IFN-γ)-induced M1 polarization of peritoneal and tumor-derived mouse macrophages [62]. The role of AXL is controversial with studies showing that induction of M1 macrophage phenotype, but also M2 macrophage signature, can be induced by AXL depending on the cytokine environment and disease context. Additionally, AXL contributes to efferocytosis in the context of inflammatory environments [63,64,65,66]. In contrast, MERTK is strongly connected to M2 macrophage polarization status and is essential for efferocytosis predominantly in anti-inflammatory environments [67,68,69,70].

If and how TAM receptors influence osteoclastogenesis and bone resorption by macrophage polarization is unknown. However, inhibition of MERTK might be an attractive treatment strategy for targeting macrophages to induce M1 polarization, which may inhibit osteoclastogenesis. Consistently, we detected reduced TRAP^+^ mononuclear cells induced by silencing MERTK genetically, as well as by targeting with a pharmacologic compound. As bone marrow macrophages express TRAP, these effects might have been mediated by altered macrophage polarization, composition, or numbers. Figure 1 summarizes the proposed mechanism of regulation of osteoclast formation by MERTK and TYRO3 (Figure 1a,b).

### 2.2. TAM Receptors in Osteoblasts

The osteogenic fate of MSCs is tightly controlled by factors present in the bone marrow microenvironment. The classical osteogenic differentiation pathways are Wnt/β-catenin and bone morphogenic protein (BMP). Furthermore, numerous hormones impact osteoblast function including insulin-like growth factor type 1 (IGF-1), parathyroid hormone (PTH), PTH-related peptide (PTHrP), active vitamin D (1,25(OH)2D3), leptin, and glucocorticoids. Other important proteins of the osteoblast signaling network are members of the fibroblast growth factor (FGF), sonic hedgehog (SHH), ephrin, and notch family [71].

The first study demonstrating that TAM receptors are potentially involved in osteogenesis showed that AXL modulates the osteogenic differentiation of pericytes [72]. Pericytes are perivascular cells that wrap around blood capillaries providing a source of undifferentiated mesenchymal cells, which can differentiate into other cell types, such as chondrocytes and osteoblasts, and induce ectopic calcification and osteogenesis [73]. The authors showed that AXL is downregulated during osteogenic differentiation. Inhibition of GAS6-AXL signaling enhanced mineralization of pericyte nodules in vitro, but the number of mineralized nodules was decreased [72]. Another study showed that AXL prevents calcified matrix deposition by vascular smooth muscle cells by inducing the PI3K-AKT pathway and inhibiting apoptotic caspase signaling [74]. In contrast, a recent study suggested that inhibition of AXL may reduce the formation of new bone, shown by slightly decreased matrix mineralization in osteoblast cultures upon AXL blockade [27]. The PI3K-AKT pathway has crucial osteoblast-specific effects, promoting proliferation, survival, and differentiation. AKT-1 knockout mice were small with reduced bone mineral density, whereas AKT-1/AKT-2 double knockouts exhibited a strong phenotype with dwarfism and negligible ossification, and died shortly after birth [75,76]. Conditional knockout of phosphatase and tensin homolog (PTEN), which negatively regulates AKT signaling, increased bone volume in mice by promoting osteoblast function and decreasing osteoblast apoptosis [77]. The role of AXL in bone development and homeostasis is largely unknown. Given its ability to increase bone nodule formation, induce the PI3K-AKT pathway, and inhibit caspase signaling, AXL might promote osteoblast differentiation by increasing proliferation and survival. Its inhibitory effect on mineralization suggests a negative regulatory role in mature osteoblast function, which should be explored in further studies.

We recently demonstrated that MERTK and TYRO3 regulate osteoblastic bone remodeling. By using osteoblast-specific knockout mouse models (Col1a1-2,3kb-cre^+^Mertk^flox/flox^ and Col1a1-2,3kb-cre^+^Tyro3^flox/flox^), we identified MERTK as a negative regulator of bone mass, whereas TYRO3 enhanced osteoblast differentiation and bone formation. Treatment of healthy mice with the MERTK-specific inhibitor R992 increased bone mass by enhancing osteoblastic bone formation. Mechanistically, we found that MERTK induces the RHOA-Rho-associated protein kinase (ROCK)-MYOSIN II pathway, whereas TYRO3 inhibits RHOA-MYOSIN II signaling. Thereby, MERTK induced cytoskeletal remodeling, which promoted cellular contraction and stress fiber formation, whereas TYRO3 stimulated opposite effects [25]. The promising role of MERTK as a target supporting osteoblasts was recently underlined by another study by Decker et al. It was demonstrated that inhibition of MERTK primed alveolar bone mesenchymal stem cells to differentiate into matrix-producing osteoblasts by activating WNT signaling [27].

The WNT/β-CATENIN pathway is the key pathway for osteoblast differentiation. Extracellular ligands of the WNT family glycoproteins (e.g., WNT1, WNT5a, WNT7b, WNT10b, and WNT16) bind to the seven-pass transmembrane G-protein-coupled receptors of the frizzled (FZD) family and its co-receptor of the arrow/low-density lipoprotein receptor-related protein (Lrp) family (e.g., LRP5 and LRP6). The WNT extracellular antagonists Dickkopf (DKK1) and Sclerostin (SOST) represent negative regulators of osteoblast differentiation and bone mass, which are emerging promising therapeutic targets for osteoanabolic therapy. The activation of canonical WNT signaling leads to stabilization and translocation of intracellular β-CATENIN into the nucleus to regulate gene transcription of lymphoid-enhancing factor/T-cell factor (LEF/TCF) to stimulate expression of osteoblast target genes. Cytoplasmic β-CATENIN is regulated by phosphorylation by a protein degradation complex composed of glycogen synthase kinase 3β (GSK3β), axis inhibition protein (AXIN), and adenomatous polyposis coli (APC). GSK3β phosphorylates and tags β-CATENIN for ubiquitination and proteasomal degradation [78].

The RHOA-ROCK pathway has recently emerged as a strong inhibitory pathway for WNT signaling, thereby negatively regulating osteoblast differentiation. It was shown that RHOA knockout in osteoblasts increases bone mass while overexpression decreases it. RHOA-ROCK activation in osteoblasts activated Janus kinase (JAK)1/2, which directly phosphorylated GSK3β, resulting in GSK3β activation and subsequent β-CATENIN destabilization, equalizing osteoblast-promoting wingless-related integration site (WNT) signaling. RHOA loss of function interacted genetically with DKK1 gain of function to rescue the severe limb truncation phenotype in mouse embryonic limb bud ectoderms [79]. Furthermore, another study showed that RHOA-ROCK inhibits IGF-1 signaling in osteoblasts. IGF-1 is one of the most abundant growth factors deposited in the bone matrix. RHOA-ROCK inhibited phosphorylation of insulin receptor substrate 1 (IRS-1) and AKT, key effectors of the IGF-1/IRS-1/PI3K/AKT pathway in osteoblasts [80].

MERTK emerges as a potential key regulator of WNT/β-CATENIN, as well as IGF-1 signaling in osteoblasts via the RHOA-ROCK pathway. As MERTK acts via ROCK downstream of key extracellular osteogenic signals, including members of the WNT family or IGF-1, MERTK could represent a suppressor of osteoblastogenesis independent of important osteoblast growth factors. It was found that the ability of RHOA-ROCK to inhibit WNT signaling is lower in physiological settings in contrast to pathological conditions like age-related bone loss [79]. Therefore, blocking MERTK might be a tool to treat osteopenic bone disorders. In contrast, TYRO3 agonists could prevent RHOA-ROCK signaling in osteoblasts, thereby improving osteoblast function and bone health. Romosozumab is approved for osteoporosis and targets Sclerostin, thereby increasing WNT signaling in osteoblasts [81,82]. As MERTK-RHOA-ROCK signaling interferes with WNT signaling downstream of Sclerostin, activation of MERTK could represent a potential resistance mechanism for Sclerostin-directed therapy in osteoporosis and other bone diseases.

Studies investigating the function of TAM receptor ligands GAS6 and PROS1 in bone homeostasis are scarce. GAS6 and PROS1 are both highly expressed by osteoblasts and PROS1 was found to inhibit osteoblastogenesis via MERTK in vitro [25]. Nevertheless, PROS1 might also promote bone formation via binding to TYRO3 in vivo. The role of GAS6 in osteoblast differentiation and function is unknown. Additional functional studies are needed to widen the knowledge regarding TAM receptor ligands as part of the osteoblast signaling network and how they might contribute to bone diseases. Figure 2 shows the possible interaction of TAM receptors with important members of the osteoblast signaling network (Figure 2).

## 3. TAM Receptors in Bone Health and Disease

### 3.1. TAM Receptors in Postmenopausal Osteoporosis

Osteoporosis is a bone disease characterized by impaired bone microarchitecture and reduced bone mineral density (BMD) leading to decreased bone strength with increased fracture risk [83]. The underlying pathomechanism consists of increased bone turnover induced by osteoclastic bone resorption and osteoblast dysfunction [84]. Postmenopausal osteoporosis is the most common type of osteoporosis and is caused by estrogen deficiency [83]. Interestingly, PROS1 as well as GAS6 expression were found to be regulated in principle by sex hormones. Estrogen was found to downregulate PROS1 expression by inhibiting PROS1 promotor activity and by upregulation of miR-494, which directly targets PROS1 [85,86]. Acquired PROS1 deficiency has been reported in individuals with high levels of estrogen during pregnancy and in those taking oral contraception, leading to a high risk for deep venous thrombosis [87,88,89]. In contrast, GAS6 gene expression was found to be induced by an estrogen response element (ERE) in the GAS6 promoter [90]. Furthermore, the Gas6 gene contains androgen-responsive elements; therefore, testosterone can directly influence Gas6 gene transcription and protein expression [91]. Serum GAS6 concentrations were positively associated with testosterone concentrations and inversely correlated with age [92,93].

Recently, a single-nucleotide polymorphism (SNP) associated with the PROS1 gene was attributed to an etiological role in osteoporosis. The study used summary statistics from large-scale genome-wide association studies (GWAS) conducted for SHBG, a protein modulating the concentration and bioavailability of sex hormones and heel-estimated BMD (eBMD), a widely accepted metric for osteoporosis management and fracture risk assessment. The authors demonstrated an inverse correlation between SHGB and eBMD, indicating that SHGB promotes osteoporosis. Furthermore, the SNP rs8178616 mapped to the PROS1 gene was found to be correlated with SHBG and eBMD, giving rise to the hypothesis that PROS1 may play a role in the pathogenesis of osteoporosis [94].

More studies should be conducted investigating the interplay of sex hormones, TAM receptor ligands, and postmenopausal osteoporosis.

### 3.2. TAM Receptors in Bone Healing

Bone regeneration is a dynamic and complex physiological process involving the interaction of multiple cell types, cytokines, and growth factors, and aims at tissue regeneration and structural reconstitution [95]. The immune system is known to be crucial for the early phase of bone healing as the repair process starts with a local inflammatory response which triggers downstream processes involving MSCs, fibroblasts, osteoblasts, osteoclasts, and endothelial cells [96]. Consistently, it has been demonstrated that bone regeneration is disturbed in certain immune disorders (i.e., rheumatoid arthritis, multiple sclerosis) or diseases with systemic inflammation (i.e., diabetes, postmenopausal osteoporosis) [97].

TAM receptors and their ligands have been shown to support resolving inflammation [98,99] and maintain bone homeostasis [25], thus they may have therapeutic benefits in bone regeneration and promote tissue repair. It was recently demonstrated that the treatment of mice with a pan-TAM inhibitor (RXDX-106), which inhibits all three TAM receptors, enhanced bone healing following tooth extraction. Moreover, specific inhibition of MERTK by MRX-2343 in alveolar bone MSCs increased osteoblast differentiation even more compared with pan-TAM-treated cells. This was confirmed using Mertk^−/−^ mice which displayed accelerated bone fill following tooth extraction compared with wild-type animals, which was shown by increased bone volume and decreased trabecular separation. FACS analysis at an early time point after tooth extraction revealed that immune cell composition was not changed. Albeit, in line with the anti-inflammatory role of MERTK in immune cells, RNAseq analysis of extraction sockets revealed highly upregulated innate immune cell markers, such as CD14, CCR1, CXCL1, and TNF, as well as gene enrichment for inflammatory pathways in Mertk^−/−^ mice. Comparison with bone differentiation markers showed upregulated genes for Jagged1, Osteopontin, and Runx2 in extraction sockets of Mertk^−/−^ mice. Interestingly, genes for Alkaline Phosphatase, Collagen1a1, and Osteocalcin were surprisingly downregulated [27].

In conclusion, this study showed that TAM receptor MERTK negatively regulates bone healing, as genetic deletion of MERTK as well as inhibition of MERTK by MRX-2343 resulted in increased mineralization and accelerated bone regeneration after injury (Figure 3). As treatment with the pan-TAM inhibitor induced similar effects, other TAM receptors could also be implicated in bone healing. Given the literature on the role of AXL and TYRO3 in osteoblasts and osteoclasts, they may also contribute to bone healing by regulating bone formation or inflammatory response. However, further studies are required to investigate the precise mechanisms regulated by TAM receptors and their ligands in bone healing.

### 3.3. TAM Receptors in Inflammatory Bone Loss in Periodontal Disease

It was shown that TAM receptors are involved in periodontitis, a chronic inflammatory dental disease caused by periodontal pathogens, such as Porphyromonas gingivalis, Treponema denticola, and Tannerella forsythia, leading to alveolar bone loss. The PROS1-TYRO3 axis protected against LPS-induced inflammation in human gingival epithelial cells via the suppressor of cytokine signaling (SOCS) 1/3 and the signal transducer and activator of transcription (STAT) 1/3. Additionally, daily subcutaneous injection of PROS1 attenuated periodontitis-mediated alveolar bone loss and osteoclastogenesis in vivo [100].

In contrast, a study that investigated age-associated periodontitis of young (2–4 months) versus aged (20–24 months) Gas6^−/−^ mice demonstrated a bone-destructive role for GAS6 in periodontitis. Despite higher inflammatory response in the gingiva of Gas6^−/−^ mice, the authors detected decreased age-associated alveolar bone loss compared with Gas6^+/+^ mice. This was corroborated by decreased gingival RANKL-osteoprotegerin (OPG) ratio in Gas6^−/−^ mice, suggesting dampened osteoclastogenesis [101].

Interestingly, in this context, both ligands dampen inflammatory responses but induce antagonistic effects on alveolar bone loss, which might be mediated by osteoclasts or non-immune cells such as mesenchymal stem cells and osteoblasts (Figure 3). The precise mechanisms and the TAM receptors involved remain largely unknown. Further studies are required to explore the role of PROS1 and GAS6 in inflammatory bone loss.

### 3.4. TAM Receptors in Rheumatoid Arthritis

Rheumatoid arthritis (RA) is a systemic autoimmune disorder characterized by abnormal synovial proliferation, primarily affecting joints causing bone and cartilage destruction, disability, and decreased quality of life [102]. The prevalence of RA is around 1% of the world’s population [103]. While there is no cure for RA, different treatment strategies can reduce pain and inflammation and protect the normal function of joints as well as prevent further progression of the disease [104]. Non-steroidal anti-inflammatory drugs, disease-modifying anti-rheumatic drugs, glucocorticoids, and biological agents are used to treat RA [104]. However, only a few drugs are both disease-modifying and tolerable for joint destruction and reduced function of joints [105]. Moreover, anti-inflammatory and analgesic drugs usually cause side effects due to their non-specific distribution [106].

Osteoclasts are a therapeutic target of RA as they contribute significantly to the occurrence and progression of RA bone destruction [107]. Several studies have shown that the synovial tissue of RA patients contains high amounts of osteoclasts but also many monocytes and macrophages, which can undergo osteoclast differentiation upon specific stimuli [107,108,109,110]. For instance, synovial cells release pro-inflammatory prostaglandins and cytokines such as TNF-α and IL-6, which promote osteoclastic bone resorption by modulating the RANK/RANKL/OPG signaling pathway [111,112,113]. TAM receptors are emerging as important regulators of osteoclast activity in RA. Xue et al. showed that CD14^+^CD16^−^ monocytes are the main source of osteoclast precursors in RA since the expression of TYRO3 on CD14^+^CD16^−^ monocytes was significantly increased in RA patients and positively correlated with disease manifestation [44]. Furthermore, TAM receptor ligand GAS6 is elevated in RA synovial tissue [114] and GAS6-mediated osteoclast differentiation of CD14^+^CD16^−^ monocytes was inhibited by anti-TYRO3 antibody in a dose-dependent manner [44]. In a mouse model of arthritis, it was shown that Tyro3^−/−^ mice displayed decreased synovial inflammation, hyperplasia, and bone erosion [42]. In contrast to the disease-promoting role of TYRO3 in inflammatory arthritis, AXL and MERTK play a protective role in RA. Waterborg et al. showed that Mertk^−/−^ mice displayed exacerbated arthritis pathology demonstrated by severe joint inflammation and bone erosion in a mouse model of RA, the KRN STA serum transfer arthritis model. The authors further showed that overexpression of PROS1 reduced inflammatory response and bone damage in knee joints, confirming the protective role of MERTK [15]. The GAS6-AXL axis prevented the release of pro-inflammatory cytokines in osteoarthritic fibroblast-like synoviocytes [115]. Another study showed that reduced levels of pro-inflammatory cytokines such as TNF-α, IL-6, and IL-1 induced by MERTK as well as AXL decreased osteoclast generation and activity, thereby ameliorating joint destruction and disease progression. In contrast, TYRO3 promoted opposite effects, fostering bone and cartilage erosion [116].

In conclusion, several studies demonstrated that TAM receptors and their ligands play a role in RA by modulating inflammatory response and osteoclast differentiation and activity (Figure 3). Thus, they are promising therapeutic targets for RA treatment.

## 4. TAM Receptors in Primary Malignant Bone Tumors and Cancer Bone Metastases

### 4.1. Osteosarcoma

Osteosarcoma is the most common primary bone cancer, with most incidence occurring in childhood and adolescents or adults older than 60 years of age. This type of malignancy accounts for approximately 0.2% of all cancers diagnosed. Osteosarcoma is clinically aggressive and often metastasizes to the lung leading to a high mortality rate [117].

The available literature indicates the role of AXL as a mediator of osteosarcoma metastasis. A study using tissue microarray from human osteosarcoma samples reported increased AXL expression compared with corresponding adjacent non-cancerous tissue. Moreover, the knockdown of AXL in a human osteosarcoma cell line induced apoptosis by increasing poly (ADP-ribose) polymerase 1 (PARP) expression and inhibited proliferation via AKT signaling [118]. Increased phosphorylation of AXL has been described to be correlated with strong expression of MMP-9, a metalloprotease modulating the metastatic cascade, as well as with recurrence of lung metastasis in osteosarcoma patients. In the same study, it was also shown that incubation of human osteosarcoma cell lines with GAS6 protected tumor cells from entering apoptosis and promoted cell migration and invasion [119]. An additional study in osteosarcoma focused on the relationship between AXL and yes-associated protein (YAP)/transcriptional coactivator with PDZ-binding motif (TAZ) proteins, the main mediators of the Hippo pathway, which is crucial for the control of organ size and reprogramming of cancer cells in metastasis [120]. YAP/TAZ can translocate into the nucleus and exert their function as transcriptional coactivators, inducing AXL expression. It was found that AXL was expressed at high levels in human metastatic osteosarcoma cell lines, as well as in derived circulating tumor cells (dCTCs), which are cancer cells shed to the blood and founders of metastases. By using a small molecular inhibitor of AXL in vivo, it was demonstrated that AXL inhibition significantly reduced the homing of the osteosarcoma cells into the lungs, diminishing the number of pulmonary metastases [120,121].

Additionally, MERTK mediates macrophage efferocytosis in osteosarcoma and induces an immunosuppressive microenvironment by inducing M2 polarization and promoting programmed death-ligand 1 (PD-L1) expression in macrophages. Treatment with a MERTK inhibitor in an in vivo model of osteosarcoma reduced tumor growth but did not affect the number of lung metastases. Analysis of the tumor microenvironment showed that MERTK blockade reverted the M2 polarization of the macrophages and enhanced the infiltration of CD8^+^ T cells and their cytotoxic function [67]. In summary, TAM receptors represent targets to suppress tumor growth and metastasis in osteosarcoma. Additionally, blocking TAM receptor signaling might induce alterations in the pro-tumorigenic osteosarcoma microenvironment favoring anti-tumor immune response, which should be the subject of further studies. Especially, the role of PROS1 and TYRO3 in osteosarcoma remains unknown.

### 4.2. Multiple Myeloma

Multiple myeloma (MM) is the second most common hematologic malignancy and is characterized by monoclonal expansion of malignant plasma cells (PCs) in the bone marrow. Diagnostic criteria include more than 10% bone marrow infiltration of PC, anemia, decreased renal function caused by expansion of malignant PCs, hypercalcemia, and lytic lesions in the bone, which is the hallmark of this disease [122]. The interaction of myeloma cells with the bone marrow microenvironment results in a vicious circle leading to osteoclast activation, osteoblast dysfunction, and subsequently osteolysis and disease progression [123].

We demonstrated that MERTK, but not AXL or TYRO3, is crucial for MM progression. GAS6 and MERTK expression were found to be upregulated in malignant bone marrow PC of myeloma patients compared with healthy donor samples [124]. Additionally, downregulation of MERTK expression or therapeutic blockade of GAS6 by Warfarin resulted in reduced disease burden and prolonged overall survival in a systemic myeloma mouse model [124]. Further investigations demonstrated that MERTK contributes to osteoblast dysfunction in myeloma bone disease. In vitro culture of MERTK knockout osteoblasts showed increased matrix mineralization in the presence of conditioned medium harvested from myeloma cells compared with osteoblasts isolated from control littermate mice expressing the receptor. This effect was induced by myeloma cells activating MERTK-MYOSIN II signaling in osteoblasts, which is detrimental to their function. Blocking MERTK by small molecule inhibitor R992 exerted therapeutic anti-myeloma effects in an orthotopic xenograft myeloma mouse model. Moreover, R992 counteracted cancer-induced bone loss and normalized bone homeostasis by increasing osteoblast differentiation and function, leading to reduced tumor progression and increased overall survival in vivo [25]. Another study demonstrated that GAS6, secreted by bone marrow stromal cells, upregulated IL-6 expression, a major growth factor of MM. Vice versa, IL-6 increased GAS6 expression in MM cells, leading to an autocrine/paracrine loop fostering MM growth. Thus, both pathways, IL-6 and GAS6-MERTK, contribute synergistically to the pathogenesis of MM [125].

A more recent study showed that major histocompatibility complex (MHC) class I chain-related protein A (MICA), a cell surface ligand for NKG2D mediating natural killer (NK) cell response, is regulated by the GAS6-TAM receptor axis in MM cells. The authors proved that GAS6, secreted by bone marrow stromal cells, activated AXL and MERTK signaling, which promoted MICA expression in myeloma cells via the NF-kB pathway [126]. Another study observed a correlation between the increased frequency of M2 macrophages expressing MERTK and elevated CXCL13 levels in the bone marrow of MM patients. It is known that CXCL13 influences the composition of the MM microenvironment, leading to enrichment in tumor-supporting immune cells and, therefore, promoting disease progression [127].

All in all, autocine and paracrine GAS6 induce MM progression via MERTK. In contrast, AXL induces MM quiescence. The role of PROS1 and TYRO3 in MM remains largely unknown. Especially, the immunomodulatory role of TAM receptors in MM cells and the immune microenvironment should be further explored to identify targets to foster ant-myeloma immune response.

### 4.3. Bone Metastasis

Cancer represents a complex group of diseases that is caused by genetic changes. It is one of the leading causes of mortality worldwide, with metastasis being the cause of death for more than 90% of cancer patients [128,129]. Most cancers metastasize to particular target organs. In this context, several tumor entities including the most common cancer types, breast, prostate, and lung cancer, oftentimes metastasize to bone [128,130]. Overall, bone metastasis reduces quality of life and patient survival. Specifically, patients with bone metastases often suffer from pain, pathologic fractures, hypercalcemia, and spinal cord compression due to increased bone resorption or intraspinal growth of metastases [130,131].

Bone metastasis can lead to osteolytic, osteoblastic, or mixed osteolytic–osteoblastic bone lesions. Osteolytic lesions are often caused by breast and lung cancer bone metastasis, in which tumor cells induce a vicious cycle of excessive osteoclastic bone resorptive activity and bone loss. Osteoblastic lesions are typically present in the prostate and sometimes in lung cancer bone metastasis. In this case, there is increased osteoblast activity resulting in sclerotic bone lesions. In mixed lesions, both lytic and osteoblastic processes coexist. This type of bone metastasis is especially recurrent in breast cancer patients [130,132].

At present, there are few published studies on the role of TAM receptors in bone metastasis. It has been shown that knockdown of AXL as well as its inhibition by Bemcentinib in prostate and breast cancer cells reduced the capacity of migration and invasion of these cells in vitro and reduced bone metastatic lesions in vivo. Thus, AXL inhibition suppressed breast and prostate bone metastasis [36].

Another study identified MERTK as a driver of bone metastasis in prostate cancer in a screening of human samples. High MERTK expression was found more frequently in prostate cancer bone metastatic tissue compared with the primary tumor. These findings were validated by in vivo models [133]. Furthermore, we demonstrated that MERTK and TYRO3 play a role in the development of osteolytic lesions in preclinical models of breast cancer metastasis. We utilized a syngeneic mouse model of bone metastasis by using the EO771 breast cancer cell line in mice harboring a conditional deletion of MERTK and TYRO3 in osteoblasts (Col1a1-2,3kb-cre^+^Mertk^flox/flox^ and Col1a1-2,3kb-cre^+^Tyro3^flox/flox^). MERTK contributed to osteoblast dysfunction in osteolytic bone disease induced by breast cancer cells, whereas TYRO3 was bone-protective by promoting osteoblastogenesis and bone formation. In addition, in vivo treatment of breast and lung cancer bone metastasis with MERTK small molecule inhibitor R992 increased bone volume and overall survival. These data showed that targeting MERTK might be an approach to treating bone metastasis-induced osteolytic bone disease by increasing osteoblast function [25]. An additional study showed that blocking MERTK might also influence osteoclastic bone resorption in breast cancer bone metastasis. By using the syngeneic EO771 breast cancer bone metastasis model, we observed increased bone volume and reduced osteoclast numbers in bones of those animals in which MERTK was not expressed in the myeloid lineage (LysM-cre^+^Mertk^flox/flox^) [26]. All in all, MERTK represents a target in osteolytic bone metastasis to decrease tumor progression and increase bone mass by stimulation of osteoblast differentiation and function and simultaneous inhibition of osteoclast formation. In contrast, TYRO3 increased bone mass in osteolytic bone metastasis by increasing osteoblastogenesis (Figure 4a). The role of TAM receptors in osteoblastic bone metastases is unknown. The potential osteoblast-stimulating role of TYRO3 and AXL raises the hypothesis that these TAM receptors might drive osteoblastic lesions in bone metastases.

### 4.4. Tumor Cell Dormancy

Bone marrow offers a perfect niche for many different tumor cells to metastasize. It comprises specific microenvironments including cavities rich in blood vessels, hematopoietic, stromal, and bone cells secreting a wide range of cytokines supporting tumor cells [134,135,136].

The cells that leave the primary tumor and reach a distant organ are called disseminated tumor cells (DTCs). DTCs in the bone can evade different cancer treatments and survive for prolonged periods in a state called dormancy before becoming reactivated to form overt metastases [137]. Thus, tumor dormancy is a reversible state controlled at least in large parts by the bone microenvironment. A key role in dormancy is attributed to osteoblasts, which can induce and maintain cancer cell dormancy. It has been shown that osteoblasts secrete growth differentiation factor 10 (GDF10) and TGFβ2, factors that induce quiescence of prostate cancer cells activating the p38MAPK/T252RB pathway [138]. Just like in bone remodeling, osteoblasts and osteoclasts exert opposite roles in regulating tumor dormancy [139]. In this context, osteoclasts have been described to reactivate multiple myeloma dormant cells. For example, the activation of osteoclasts by RANKL induces bone resorption, a process that can reactivate and release dormant tumor cells from the endosteal niche [140]. There is evidence that TAM receptors can function as a molecular switch between dormant and proliferative states in prostate cancer cells depending on cell-intrinsic TAM receptor expression. In preclinical models of prostate cancer, tumor cell homing was significantly higher in the hind limb than the fore limb bones, which was negatively correlated with levels of secreted GAS6 in these bones [141]. Furthermore, GAS6 expression and its interaction with TAM receptors modulated the dormant status of prostate tumor cells. Indeed, once prostate cancer cells reach the osteoblastic niche, secretion of GAS6 by osteoblasts is upregulated, inducing growth arrest in cancer cells [142,143]. This mechanism was previously described in a study investigating the maintenance of HSCs by bone marrow stromal cells via GAS6 [144]. In addition, GAS6 was able to regulate dormancy depending on the expression levels between AXL and TYRO3 in DTCs. When TYRO3 expression was higher than AXL, tumor cells proliferated and, on the other hand, when AXL expression levels exceeded those of TYRO3, the cancer cells remained largely quiescent [143]. Another study supporting these data showed that prostate cancer cells, in which AXL expression was downmodulated, exhibited a suppressed cellular dormancy phenotype in the bone marrow compared with control transduced cells in vivo. Mechanistically, the AXL-GAS6 axis induced expression of TGF-β ligand and TGFBR2, which was reverted by the knockdown of AXL [145,146]. Most recently, it has been published that AXL was one of the genes enriched in a transcriptome signature of dormant myeloma cells. Co-cultures of murine myeloma cells with osteoblasts upregulated AXL expression in myeloma PCs. Moreover, the blockade of AXL using cabozantinib and BMS-777607 in an in vivo myeloma model decreased the numbers of dormant myeloma cells, allowing them to proliferate and promote disease progression [37].

In prostate cancer, MERTK has also been described as a regulator of tumor cell dormancy. Authors have shown that the knockdown of MERTK in prostate cancer cells induced transcription factors that promote cancer dormancy, such as SOX2, NR2F1, and NANOG [147]. Loss of MERTK expression was also associated with increased p27, a cell cycle inhibitor, and diminished activation of the p38 pathway, a marker of dormant phenotype. Moreover, MERTK knockdown was driving tumor cells into the cell cycle G0/G1 phase. Most importantly, the depletion of MERTK in prostate cancer cells and subsequent inoculation in vivo increased metastasis-free survival without affecting the growth kinetics of the primary tumor [148].

TAM receptors might play a key role in the regulation of cancer cell dormancy. It was demonstrated that GAS6, which can bind to all three TAM receptors, is secreted by bone marrow stromal cells, especially osteoblasts, and induces dormancy in several cancer entities via AXL. Upon activation, the different TAM receptors can fulfill antagonistic roles in the same cell type in the regulation of proliferation, thereby representing a molecular switch for quiescence and reactivation of dormant cancer cells. In particular, activation of MERTK induces dormancy escape and bone metastasis progression (Figure 4b).

## 5. Conclusions and Future Perspectives

So far, research has focused on the TAM receptor’s prominent regulatory role in the immune, reproductive, hematopoietic, vascular, and nervous systems. Due to recent advances in the investigation of TAM receptors in the bone field, a new chapter can be opened for suitable therapeutic targets for various bone diseases. TAM receptors directly influence osteoblast and osteoclast biology with positive and negative regulatory functions, showing a highly regulated and complex biological system in both cell types. To develop a deeper understanding of the therapeutic effects observed by TAM receptor blockade, more studies need to be conducted to investigate the role of TAM receptors and their ligands, GAS6 and PROS1, in the regulation of different biological functions, such as proliferation, differentiation, and migration of osteoblasts and osteoclasts. The influence of sex and aging on TAM receptor signaling needs to be explored more robustly and underlines their possible promising role in the pathogenesis of bone diseases such as osteoporosis. Additionally, the combination of tumor-promoting effects in many different cancer entities with regulatory effects in osteoblasts and osteoclasts leave TAM receptors as promising therapeutic targets for primary malignant bone tumors, metastatic bone disease, and dormancy reactivation.

## Figures and Tables

**Figure 1 ijms-25-00233-f001:**
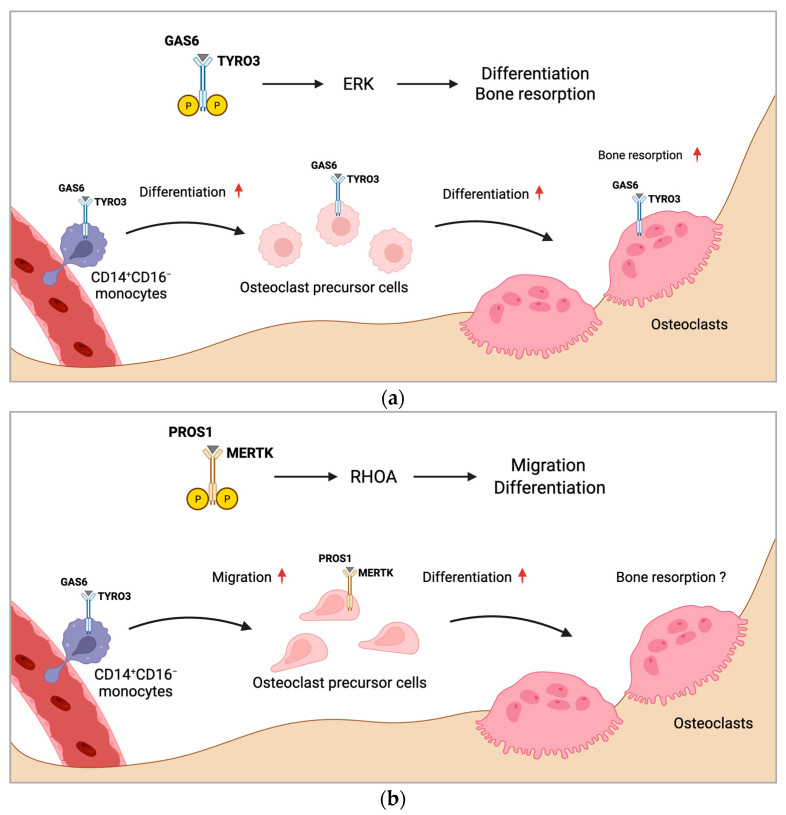
Proposed mechanism of regulation of osteoclast formation by MERTK and TYRO3. (**a**) CD14^+^CD16^−^ monocytes with increased TYRO3 expression are the main source of osteoclast precursors. GAS6 induces TYRO3 phosphorylation and ERK signaling, promoting osteoclast differentiation. Activation of TYRO3 increases bone resorptive activity of osteoclasts. (**b**) MERTK induces RHOA signaling in osteoclast precursor cells leading to enhanced osteoclast differentiation and migration. Rounded arrows indicate differentiation steps in osteoclastogenesis. Straight horizontal arrows indicate TAM receptor signaling pathway leading to different biological functions in osteoclastogenesis. Straight vertical arrows show upregulation of indicated biological functions. The role of MERTK in osteoclastic bone resorption is unknown (?). Created with BioRender.com.

**Figure 2 ijms-25-00233-f002:**
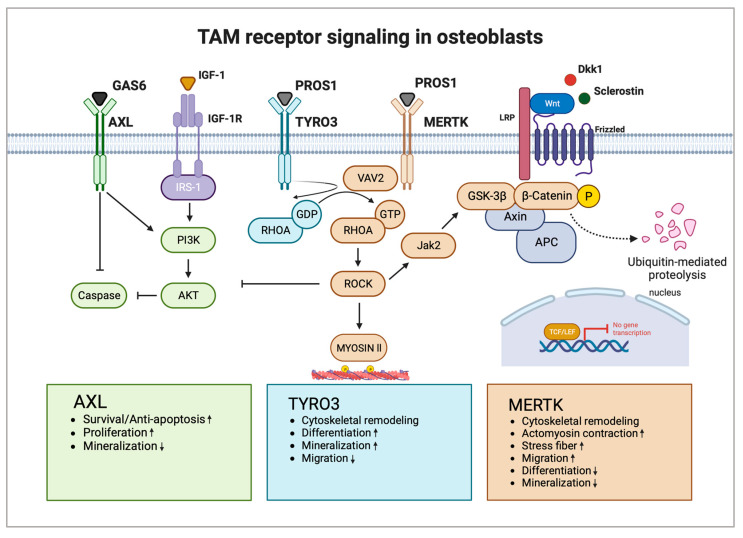
TAM receptors and their downstream signaling pathways in osteoblasts. AXL induces osteoblast proliferation and survival via activation of the PI3K-AKT pathway and inhibition of caspase signaling but inhibits osteoblast mineralization. TYRO3 inhibits RHOA signaling in osteoblasts, promoting cytoskeletal remodeling leading to enhanced differentiation and mineralization but reduced migration. MERTK induces RHOA-ROCK signaling in osteoblasts, which negatively regulates important osteoblastogenesis pathways WNT and IGF-1. Activation of MERTK negatively regulates differentiation and mineralization but promotes migration. Main pathways and biological functions induced by AXL are coloured in green, TYRO3 in turquoise and MERTK in brown. Arrows illustrate activation and blunt arrows silencing of indicated pathways. Boxes at the bottom highlight biological functions of AXL, TYRO3 and MERTK in osteoblasts. Arrow up: Increased function; Arrow down: Decreased function. IGF-1R: IGF-1 receptor; GDP: guanosine diphosphate; and GTP: guanosine triphosphate. Created with BioRender.com.

**Figure 3 ijms-25-00233-f003:**
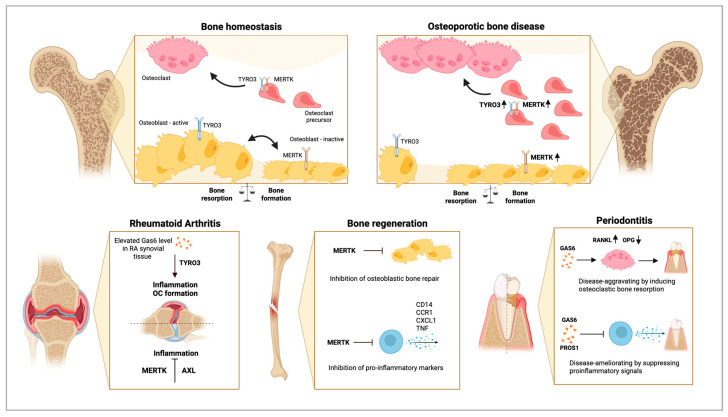
Role of TAM receptors in bone health and disease. In bone homeostasis, bone formation and bone resorption are coupled and tightly regulated. In bone-forming osteoblasts, the TAM receptor TYRO3 induces bone formation, whereas activation of MERTK limits osteoblastic bone-forming capacity. Additionally, MERTK and TYRO3 promote osteoclastogenesis. In osteoporotic bone diseases, MERTK and TYRO3 induce excessive osteoclast formation and bone resorption. Activation of MERTK leads to osteoblast dysfunction and inhibition of bone formation. In rheumatoid arthritis, TYRO3 induces proinflammatory signaling, aggravating the disease. MERTK and AXL induce anti-inflammatory pathways, limiting disease progression. MERTK plays a negative role in bone regeneration by limiting osteoblastic bone repair and pro-inflammatory response to allow immune cell infiltration for the initiation of bone healing. In periodontitis, PROS1 and GAS6 suppress proinflammatory signaling, ameliorating the disease. Albeit GAS6 can induce osteoclastic bone resorption by increasing the RANKL/OPG ratio, which is disease aggravating. Rounded arrows indicate differentiation steps in osteoclastogenesis and osteoblastogenesis. Straight arrows and blunt arrows indicate activation and blocking, respectively, of different biological functions and disease states. Arrow up: Increased RANKL; Arrow down: Decreased OPG. Created with BioRender.com.

**Figure 4 ijms-25-00233-f004:**
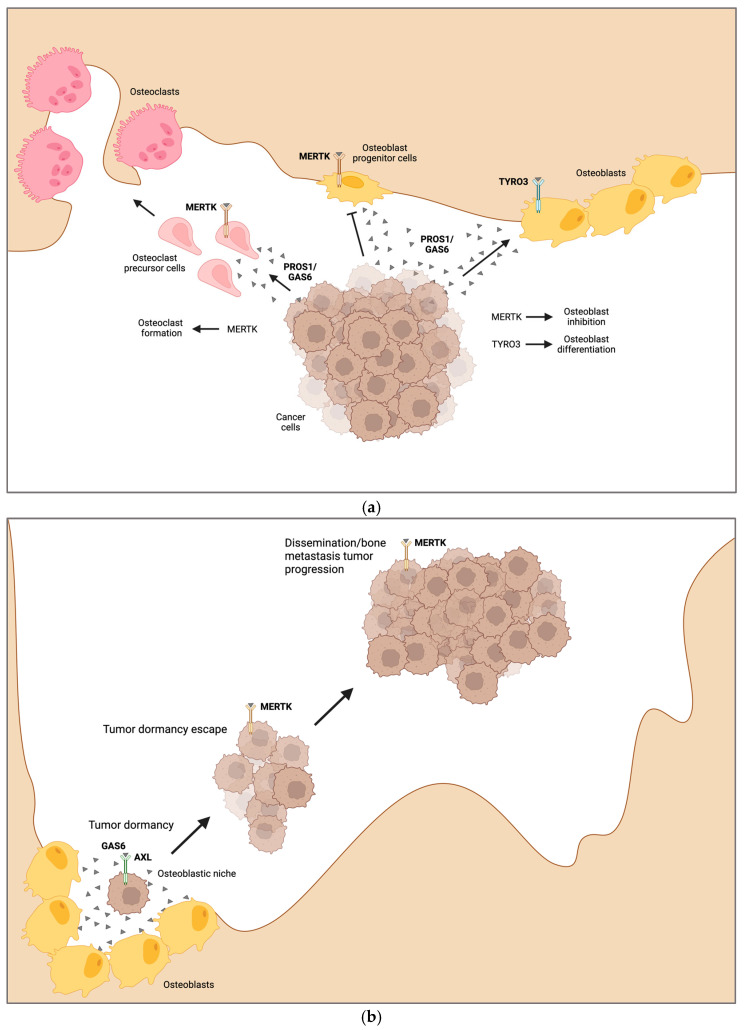
Role of TAM receptors in tumor cell dormancy and bone metastatic disease. (**a**) In bone metastatic disease of breast or lung cancer and myeloma bone disease, cancer cells activate MERTK by secreting PROS1 and GAS6 to inhibit osteoblastogenesis and promote osteoclast precursor cell differentiation and migration to resorption sites to form osteoclasts leading to osteolysis. TYRO3 is bone protective in bone metastases by increasing osteoblast differentiation, which may lead to osteoblastic lesions. (**b**) Osteoblasts secrete high levels of GAS6, inducing tumor cell dormancy via AXL in prostate cancer and multiple myeloma. Unknown mechanisms lead to the upregulation of MERTK expression inducing proliferation and dormancy escape, which promotes tumor dissemination and progression. Arrows and blunt arrows indicate activation and blocking, respectively, of different cancer-related pathological processes in the tumor microenvironment in the bone. Created with BioRender.com.

## Data Availability

Not applicable.

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
