# Peer review of "The Role of TAM Receptors in Bone"

_ijms, 2023, doi:10.3390/ijms25010233_

Round 1

Reviewer 1 Report

Comments and Suggestions for Authors

Analysing the literature database, it is clear the importance of adequately understanding the role of different receptors / regulators in the onset and progression of various diseases. Therefore, the present review highlights the involvement of TAM family in maintaining organ integrity and self-renewal. Furthermore, as you emphasised, these receptors play a major role in promoting tumour progression, metastasis and therapy resistance, and I believe your article may offer important new insights regarding these mechanisms. The review describes in detail all these aspects, the included data are updated and the figures clearly summarise the presented information. After carefully reading your work, I agree that TAM family can represent a promising therapeutic target for primary malignant bone tumours, metastatic bone disease and dormancy reactivation. I only have some few remarks regarding the used abbreviations: all the abbreviations should be explained in the abstract and manuscript, the first time they have been used. A similar suggestion regarding the abbreviations presented in the figures - they should be explained below each figure.

Author Response

General remark: Authors would like to thank all reviewers for the effort they put in reviewing this paper and for their positive evaluation of the manuscript. As requested, we have addressed the concerns and suggestions of the referees in a point-by-point response. Appropriate changes were made and highlighted in the revised manuscript according to the suggestions of reviewers.

RESPONSE TO REVIEWER #1

1) Analysing the literature database, it is clear the importance of adequately understanding the role of different receptors / regulators in the onset and progression of various diseases. Therefore, the present review highlights the involvement of TAM family in maintaining organ integrity and self-renewal. Furthermore, as you emphasised, these receptors play a major role in promoting tumour progression, metastasis and therapy resistance, and I believe your article may offer important new insights regarding these mechanisms. The review describes in detail all these aspects, the included data are updated and the figures clearly summarise the presented information. After carefully reading your work, I agree that TAM family can represent a promising therapeutic target for primary malignant bone tumours, metastatic bone disease and dormancy reactivation. I only have some few remarks regarding the used abbreviations: all the abbreviations should be explained in the abstract and manuscript, the first time they have been used. A similar suggestion regarding the abbreviations presented in the figures - they should be explained below each figure.

Reply: Authors would like to thank the reviewer for insightful comments and the positive feedback. To address the concern about the abbreviations, we have revised the text by incorporating the full form of abbreviations upon their first use in the manuscript. We have ensured that all abbreviations in the text and in the figure legends are now included with their full form.

RESPONSE TO REVIEWER #2

1) This review provides excellent overview on the role of TAM family of receptors and will be of interest to the readers of International Journal of Molecular Sciences. 

The figures in the paper are of great quality. However, I would suggest to add 1-2 figures related to Section 3 - TAM receptors in bone health and disease. 

Reply: Authors would like to thank the reviewer for the positive feedback and the suggestion. We have addressed it by including an additional figure (Figure 3) for the role of TAM receptor in bone health and disease (Section 3, page 11).

Reviewer 2 Report

Comments and Suggestions for Authors

This review provides excellent overview on the role of TAM family of receptors and will be of interest to the readers of International Journal of Molecular Sciences. 

The figures in the paper are of great quality. However, I would suggest to add 1-2 figures related to Section 3 - TAM receptors in bone health and disease. 

Author Response

(The authors gave the same response as above.)
